# Inhibition of CCl$_4$-induced liver inflammation and fibrosis by a NEU3 inhibitor

**Darrell Pilling** [ID]*, **Trevor C. Martinez** [ID], **Richard H. Gomer** [ID]*

Department of Biology, Texas A&M University, College Station, Texas, United States of America

* dpilling@bio.tamu.edu (DP); rgomer@tamu.edu (RHG)

**Data Availability Statement:** All relevant data are within the manuscript and its Supporting information files.

## Abstract

Sialic acids are located on the ends of many glycoconjugates and are cleaved off by enzymes called sialidases (neuraminidases). Upregulation of neuraminidase 3 (NEU3) is associated with intestinal inflammation and colitis, neuroinflammation, and lung fibrosis. Genetic ablation of NEU3 or pharmacological inhibition of NEU3 reduces lung fibrosis in mice. To determine if inhibiting NEU3 can inhibit liver fibrosis in the commonly-used CCl$_4$ model, in this report, we examined the effects of injections of the NEU3 inhibitor 2-acetyl pyridine (2AP). 2AP inhibited CCl$_4$-induced weight loss in female but not male mice. 2AP attenuated CCl$_4$-induced liver inflammation and fibrosis in male and female mice, but did not affect CCl$_4$-induced steatosis. After CCl$_4$ treatment, female but not male mice had significant increases in liver neutrophils, and 2AP attenuated this response. 2AP also reversed CCl$_4$-induced liver desialylation and CCl$_4$-induced increased expression of NEU3. Patients with pulmonary fibrosis have increased desialylation of some serum proteins, and elevated serum levels of NEU3. We find that sera from patients with nonalcoholic fatty liver disease (NAFLD) and nonalcoholic steatohepatitis (NASH) have elevated desialylation of a serum protein and patients with NAFLD have increased levels of NEU3. These data suggest that elevated levels of NEU3 may be associated with liver inflammation and fibrosis, and that in mice this is ameliorated by injections of a NEU3 inhibitor.

## Introduction

An estimated 800 million people are affected by chronic liver disease [1]. In the US, over 30 million people have some form of liver disease, with over 750,000 hospitalizations and 56,000 deaths per year [2]. Liver fibrosis can occur after many forms of liver injury including trauma, hypoxia, infection, and steatosis, leading to inflammation, and then fibrosis, cirrhosis, hepatocellular carcinoma, and liver failure [1–4]. Liver fibrosis involves the accumulation of extracellular matrix (ECM) molecules which promote increased tissue stiffness and the activation of hepatic stellate cells (HSCs) which drives dysregulated liver metabolism and further liver injury [5]. The only US Food and Drug Administration (FDA) approved drug for the treatment of non-cirrhotic non-alcoholic steatohepatitis (NASH) with moderate to advanced liver fibrosis is a thyroid hormone receptor beta (TR-β) agonist [6]. Unfortunately, this drug is only effective in 30% of patients at reducing steatohepatitis (steatosis with inflammation), and only

**Funding:** This work was supported by a Beckman Scholars Program award to TM (RHG mentor) by the Arnold and Mabel Beckman Foundation (http://dx.doi.org/10.13039/100000997). The funder had no role in study design, data collection and analysis, decision to publish, or preparation of the manuscript. The authors received no other specific funding for this work.

**Competing interests:** RHG is a scientific founder of Prosia Therapies, an early-stage company developing NEU3 inhibitors as therapeutics for pulmonary fibrosis. Texas A&M University has published patent applications on the use of sialidase inhibitors (United States Patent Application 20190201485) to regulate fibrosis. DP and RHG are inventors on pending patent applications for the use of sialidase inhibitors as anti-inflammatory, anti-fibrotic, and/or anti-obesity compounds.

effective in 25% of patients at reducing fibrosis [6]. We thus need to understand if there are additional mechanisms that drive liver fibrosis where inhibiting the mechanism could potentially attenuate fibrosis.

Many mammalian proteins and some lipids are glycosylated, and many of the glycosylated structures have sialic acid at the distal end of the polysaccharide chain [7]. Sialidases, also called neuraminidases, remove the terminal sialic acid from these glycoconjugates [8, 9]. There are four known mammalian sialidases, NEU1, NEU2, NEU3, and NEU4 [8, 9]. Of the four sialidases, only NEU3 is expressed on the extracellular membrane [9–11]. Upregulation of NEU3 protein is associated with multiple diseases including intestinal inflammation and colitis, neuroinflammation, and lung fibrosis [9, 12–16]. For pulmonary fibrosis, a positive feedback loop where elevated NEU3 causes upregulation of the profibrotic extracellular signals TGF-β1 and IL-6, and TGF-β1 and IL-6 upregulate NEU3, appears to participate in fibrosis [14, 16, 17].

The standard assay to measure sialidase activity is the cleavage of the fluorogenic substrate 4-Methylumbelliferyl α-D-N-acetylneuraminic acid (4MU-NANA) [18]. All four mammalian sialidases can cleave 4MU-NANA when assayed at pH 4.5–5.5 [8, 19]. Using 4MU-NANA as a substrate, the general sialidase inhibitor N-acetyl-2,3-dehydro-2-deoxyneuraminic acid (DANA) inhibits all 4 mammalian sialidases in the 40–140 μM range [20]. We have found that the release of active TGF-β1 from human latent TGF-β1 is facilitated by the removal of sialic acids by NEU3 at pH 6.9 [14], which is within the range of extracellular pH found in the lungs of fibrosis patients [21]. The release of active TGF-β1 from human latent TGF-β1 by human and mouse recombinant NEU3 is inhibited by the NEU3 inhibitor 2-acetyl pyridine (2AP) with $IC_{50}$s of 40 ± 6 nM and 800 ± 170 nM respectively, whereas the general sialidase inhibitor DANA inhibits active TGF-β1 release at > 100 μM [14].

We previously found that injections of the general sialidase inhibitor DANA [16], injections of the NEU3 inhibitor 2AP [14], or the lack of NEU3 [15] attenuate bleomycin-induced pulmonary fibrosis in male mice. In male mice, injections of DANA or the lack of NEU3 also attenuate high fat diet (HFD)-induced liver inflammation [22], and DANA inhibits methionine and choline-deficient (MCD) diet-induced liver fibrosis [23], indicating that NEU3 regulates liver inflammation in male mice. Injections of DANA also inhibited HFD induced weight gain and steatosis, whereas $NEU3^{-/-}$ knock out mice were not resistant to HFD-induced weight gain and steatosis [22], suggesting that NEU3 does not regulate HFD-induced weight gain and steatosis.

Many human diseases and associated animal models have sex differences in disease incidence, prevalence, severity, and survival [24–26]. $CCl_4$-induced liver injury leads to inflammation, with a more pronounced inflammatory response in male mice [27–29]. We also observed that compared to male mice, female mice may have a reduced sensitivity to inhaled recombinant NEU3, and male and female mice have different expression patterns of NEU3 expression in the bleomycin-induced lung injury model [30, 30].

The HFD and short-term MCD models only cause a small amount of fibrosis [32]. A more severe level of fibrosis in mice is generated by repeated injections of $CCl_4$ [32–34]. To further determine if NEU3 is part of an additional mechanism that drives liver fibrosis, in this report, we examined if inhibiting NEU3 can inhibit fibrosis in the $CCl_4$ model in male and female mice.

## Materials and methods

### Mouse model of liver inflammation and fibrosis

To induce inflammation and fibrosis, 7–8 week old 20–25 g male and female C57BL/6J mice (Jackson Laboratories, Bar Harbor, ME) were given twice weekly intraperitoneal injections of

carbon tetrachloride (CCl$_4$; #289116, Sigma, St. Louis, MO) in 0.05 mL corn oil (#C8267, Sigma) at 0.5 mL/kg, or corn oil alone for 6 weeks, as previously described [32, 34]. At 21 days after CCl$_4$ or oil alone treatment had commenced, mice were also given daily intraperitoneal injections of 100 μl of PBS or 1 mg/kg 2-acetyl pyridine (2AP; A3029171, Ambeed, Arlington Heights, IL) in 100 μl of PBS. The corn oil or CCl4/ corn oil injections continued during the PBS and PBS/2AP treatments. All the mice were monitored daily to observe any sign of distress. Animals were housed with a 12-hour/12-hour light-dark cycle with free access to food and water, and all procedures were performed between 09:00 and noon. Mice were euthanized by CO$_2$ inhalation at day 42. Mice were randomly assigned to control and treatment groups by personnel uninvolved with the study.

## Ethics statement

This study was carried out in strict accordance with the recommendations in the Guide for the Care and Use of Laboratory Animals of the National Institutes of Health. The protocols were approved by the Texas A&M University Animal Use and Care Committee (IACUC 2020–0272 and 2023–0242). All procedures were performed under 4% isoflurane in oxygen anesthesia, and all efforts were made to minimize suffering.

## Tissue processing

Following euthanasia, the liver, heart, kidneys, spleen, lungs, gonadal white fat tissue, and interscapular brown fat tissue were removed and weighed before processing. Pieces of liver tissue were snap frozen in liquid nitrogen and stored at -80˚C; embedded in OCT compound (VWR, Radnor, PA), frozen, and stored at -80˚C; or for paraffin sections fixed in Zn-buffered formalin (0.1% ZnSO$_4$, #4382 VWR; 10% v/v of 38% formaldehyde, #10790–708 VWR) in water for 2 days at 4˚C, and then placed in 10% and then 30% sucrose (#8360–06, VWR) in PBS for 2 days each on ice. Fixed tissues were kept in 70% ethanol at room temperature until paraffin processing and sectioning at 5 μm.

## Hydroxyproline assays

Liver tissue hydroxyproline levels were measured using a Total Collagen/Hydroxyproline Assay Kit (702440; Cayman Chemicals, Ann Arbor, MI) to determine fibrosis [15], following the manufacturer's instructions. Briefly, snap-frozen liver tissue was weighed and then homogenized in water (50 mg per 0.5 mL) with a microtube pestle. Aliquots of 100 μl were mixed with 100 μl of 10N NaOH and then heated to 120˚C for 1 hour. Samples were neutralized with 100 μl of 10N HCl and stored at -20˚C. Samples were then assayed following the manufacturer's instructions.

## Histology and antibody staining

To determine the amount of steatosis (accumulation of fat in the cells of the liver), OCT embedded unfixed liver tissue sections were stained with oil red O to detect the accumulation of lipids, as described previously [22, 23, 35]. Paraffin embedded tissue sections were stained with hematoxylin and eosin (TAMU Histology core RRID:SCR_ 022201) to determine inflammation, and with Sirius red to determine fibrosis, as described previously [30, 36]. For Sirius red staining, paraffin-embedded sections were dewaxed with xylene, then rehydrated through a graded series of alcohols, and then distilled water. Slides were then incubated for 2 minutes in 0.2% phosphomolybdic acid (# 26357–01, Sigma), rinsed twice in water, and then incubated for 30 minutes at room temperature with a saturated solution of picric acid in distilled water

containing 0.1% Sirius red (#09400–10, Polysciences, Warrington, PA). Sections were rinsed in acidified water (0.5% glacial acetic acid in water) for 30 seconds and then repeatedly rinsed with distilled water, as described previously [37, 38].

For immunohistochemistry, formalin-fixed, paraffin-embedded slides were dewaxed with xylene, then rehydrated through a graded series of ethanol and distilled water [22, 23]. Antigens were retrieved using antigen-unmasking solution, as described previously [23]. Briefly, slides were incubated at 98˚C for 20 minutes in 10 mM sodium citrate pH 6 (# H-3300, Vector Laboratories, Newark, CA) for all antibodies except NEU3 which used 10 mM TRIS/1 mM EDTA pH 9. Slides were then left to cool to room temperature for 20 minutes, and then incubated for 5 minutes with 2 changes of water, and then 2 changes of PBS [22, 23, 36]. Slides were then blocked by incubation in PBS containing 2% BSA (PBS-2%BSA) for 60 minutes. Endogenous biotin was blocked by the addition of streptavidin and biotin solutions using a Streptavidin/Biotin Blocking Kit following the manufacturer's instructions (SP2002, Vector Laboratories). Slides were then incubated overnight at 4˚C with 1 µg/mL primary antibodies in PBS-2%BSA. Primary antibodies were anti-Mac2 (rat IgG2a, clone M3/38, BioLegend, San Diego, CA) to detect recruited and tissue macrophages, anti-F4/80 (rabbit mAb, D2S9R, Cell Signaling Technology, Danvers, MA) to detect tissue resident macrophages, anti-CD3 (rabbit clone SP7, NB600-1441, Novus Biologicals, Centennial, CO, USA) to detect T-cells, anti-CLEC4F (goat Ab, AF2784, Novus Biologicals) to specifically detect Kupffer cells, anti-MRP8 (goat Ab, AF3059, Novus Biologicals) to detect neutrophils, and anti-NEU3 (rabbit IgG, 27879-1-AP, Proteintech, Rosemont, IL) to detect NEU3. After washing with 6 changes of PBS over 30 minutes, primary antibodies were detected with 1 µg/mL biotinylated donkey F(ab′)2 anti-rabbit IgG (711-066-152; Jackson ImmunoResearch), biotinylated donkey anti-rat (NBP1-75379, Novus Biologicals), or biotinylated donkey F(ab′)2 anti-goat (705-066-147, Jackson ImmunoResearch) in PBS-2%BSA for 30 minutes. Biotinylated antibodies were detected by a 1:500 dilution of streptavidin conjugated alkaline phosphatase (SA-5100-1, Vector Laboratories) in PBS-2%BSA for 30 minutes. Staining was developed with the Vector Red Alkaline Phosphatase Kit (Vector Laboratories). Sections were then counterstained for 30 seconds with Gill's hematoxylin 3 (Sigma). Slides were mounted with VectaMount (Vector Laboratories).

For biotinylated lectin staining, paraffin sections were dewaxed with xylene, then rehydrated through a graded series of alcohols, distilled water, and then PBS, as described above. Endogenous biotin was then blocked as described above. Slides were then incubated with 1x Carbo-Free Blocking Solution (#SP-50400, Vector Laboratories) for 30 minutes to block endogenous lectins. Sections were then incubated in 1 µg/ml of primary biotinylated peanut agglutinin (PNA; #B-1075) or ricinus communis agglutinin (RCA; B-1085-1) both from Vector Laboratories in 1x Carbo-Free Blocking Solution overnight at 4˚C. After washing, biotinylated lectins were detected by a 1:500 dilution of streptavidin conjugated alkaline phosphatase (SA-5100-1, Vector Laboratories) in 1x Carbo-Free Blocking Solution for 30 minutes. Staining was developed with the Vector Red Alkaline Phosphatase Kit, then counterstained with hematoxylin, and mounted as described above.

## Patient samples

Serum samples were from de-identified liver patients from the NIH NIDDK liver study (NAFLD Adult: DOI: 10.58020/53bk-jk73) and with the approval of the Texas A&M University Institutional Review Board (IRB2022-1351). Serum samples were received from the NIH on 7 February 2023. Liver disease was assessed by NAFLD activity score (NAS) [2, 39] (S1 Table). Serum samples from de-identified age-matched apparently healthy controls were

collected at the Yale School of Medicine with approval from the Yale Institutional Review Board (HIC#0706002766) and with written consent from the donors, as described previously [40]. Samples were received at TAMU on 29 August 2019. Young control sera were collected between January and February 2023 at Texas A&M University with the approval of the Texas A&M University Institutional Review Board (IRB2017-0792D). All serum samples were stored at −80˚C. The authors had no access to information that could identify individual participants during or after data collection.

## Western blotting

Serum samples from controls and patients were thawed overnight at 4˚C. Serum samples were diluted 1/100 (v/v) in 20 mM sodium phosphate pH 7.4. The protein concentrations in the diluted sera were measured by OD 280/260 with a Synergy Mx plate reader (BioTek, Winooski, VT) with a Take3 Multi-Volume Plate insert and Gen5 Take3 software module and reading at 260, 280, and 320 nm to correct for nucleic acids and particulate matter, with 20 mM sodium phosphate pH 7.4 as a blank. Samples of diluted sera were mixed with 2x Laemmli sample buffer, heated to 98˚C for 10 minutes, and then stored at -20˚C. 10 µl of the heated samples were loaded on to 4%–20% Tris–glycine mini-Protean TGX gels (Bio-Rad, Hercules, CA), and samples were run at 45V for approximately 2 hours at room temperature. Samples were transferred to PVDF membranes (Immobilon P, MilliporeSigma, Burlington, MA), in Tris/glycine/SDS buffer containing 20% methanol, as described previously [40]. Blots were then blocked overnight at 4˚C with 1x Carbo-Free Blocking Solution (Vector Laboratories). Blots were then incubated with 1 µg/ ml biotinylated PNA or biotinylated RCA (both from Vector Laboratories) for 2 hours at room temperature in 1x Carbo-Free Blocking Solution. Blots were then washed with 6 changes of PBS containing 0.05% Tween-20 (VWR) over 30 minutes. Lectins were detected with streptavidin-HRP (016-030-084, Jackson ImmunoResearch, West Grove, PA), as described previously [16, 40]. SuperSignal West Pico Chemiluminescence Substrate (Thermo Scientific) was used following the manufacturer's protocol to visualize the peroxidase using a ChemiDoc XRS + System (Bio-Rad, Hercules, CA). PNA and RCA bands were measured by densitometry using the Bio-Rad Image lab software.

Serum samples from controls and patients were also assessed for NEU3, as described previously [16]. Electrophoresis and blotting was performed as described above and membranes were then blocked with Tris-buffered saline (TBS, 20 mM Tris #0497 VWR (pH adjusted to 7.4 with HCl); 140 mM NaCl #BDH9286 VWR) containing 0.05% Tween-20 (#0777 VWR) (TBS-T) and 5% non-fat milk protein (BD-Difco, Sparks, MD) for 30 minutes at room temperature. Western blots were then incubated overnight at 4˚C with 1 µg/mL anti-NEU3 antibodies (Rabbit; # 27879-1-AP, Proteintech) in TBS-T+5% milk, as described previously [16]. Following washes in TBS-T, blots were incubated with 1 µg/mL peroxidase-conjugated donkey F(ab′)2 anti-rabbit (711-036-152, Jackson ImmunoResearch). Blots were visualized as above.

## Image quantification

Tissue sections stained with antibodies, lectins, hematoxylin and eosin, picrosirius red, or oil red O were imaged with either an Amscope T670 (United Scope LLC, Irvine CA) or a Nikon Eclipse Ti2 microscope (Nikon Instruments, Melville, NY) and analyzed with ImageJ2 software Fiji version 1.53f (NIH, Bethesda, MD; https://imagej.net/Welcome) [41]. For the quantification of the number of cells around liver vessels, at least 100 vessels per H&E stained section were assessed for the accumulation of immune cells. We defined a vessel as having an accumulation of cells when there were ≥5 cells per vessel. The number of positively stained cells, and the percentage area of stained tissue, were quantified as described previously [16, 22, 23, 36].

To calculate the percentage area of stained tissue, the threshold level of staining intensity was set using positive and negative (no primary antibody) stained sections and then the threshold values were kept the same for analyzing each set of images, as described previously [15]. The total area of the image and the area stained (threshold) as a percentage of the total area of the image were then determined using ImageJ (S1 Fig). Image quantification and assessment of staining was performed in a blinded manner. No data points or subjects were excluded from analysis.

## Screening 2AP for activity in biochemical and functional assays

CYP450 enzymes and a panel of receptors were assessed for inhibition by 10 μM 2AP using HitProfilingScreen + CYP450 LeadHunter Panel at Eurofins (Eurofins Panlabs Discovery Services, New Taipei City, Taiwan). Briefly, human recombinant CYP450 enzymes were preincubated at 37˚C for 15 minutes with 10 μM 2AP. Enzyme activity was assessed by cleavage of the CYP450 fluorogenic substrates dibenzyl fluorescein (aromatase / CYP19A1, CYP2C8), 3-Cyano-7-ethoxycoumarin (CYP1A2, CYP2B6, CYP2C19, CYP2C9, and CYP2D6), or 7-Benzyloxy-4-(trifluoromethyl)-coumarin (CYP3A4), as described previously [42–45]. Inhibition of receptor binding was assessed using a panel of receptors expressed on cell lines or using tissue extracts and preincubated at 25˚C for 30–120 minutes with 10 μM 2AP. Cells or tissue extracts were then incubated with radiolabeled specific ligands a described (https://www.eurofinsdiscovery.com/catalog/hitprofilingscreen-cyp450-leadhunter-panel-tw/PP115) and inhibition of binding was reported (S2 Table).

## Statistical analysis

Statistical analysis was performed using Prism 7.05 (GraphPad Software, La Jolla, CA). Statistical significance between two groups was determined by t test, or between multiple groups using either one- or two-way ANOVA with Dunnett's or Bonferroni's post-test as indicated in figure legends, and significance was defined as $p < 0.05$.

## Results

### The NEU3 inhibitor 2-acetylpyridine (2AP) attenuates $CCl_4$-induced weight loss in mice

To determine whether the NEU3 inhibitor 2AP modulates $CCl_4$-induced liver inflammation and fibrosis, male and female C57BL/6 mice were treated with injections of oil or $CCl_4$ diluted in oil twice a week for 6 weeks. As previously observed [34], compared to control mice, $CCl_4$-treated mice had a significant drop in body weight 24–48 hours after $CCl_4$ injections with a recovery in weight over the next 2–3 days (S2A Fig), with no significant differences in weight loss between male and female mice (S2B–S2E Fig). This acute reversible weight loss may be due to a general toxicity of $CCl_4$. Starting at day 21, mice were also given daily intraperitoneal injections of 1 mg/kg 2AP or buffer (Fig 1A). Mice that received oil (the $CCl_4$ diluent), or oil with 2AP, gradually increased weight over the 42 days, indicating that 2AP has no overt effects on the mice (Fig 1A–1C and S2A–S2C Fig). Comparing $CCl_4$ and $CCl_4$+2AP treated mice, there was less weight loss in the $CCl_4$+2AP treated mice at days 39 and 40 (Fig 1A). This difference in body weights between $CCl_4$ and $CCl_4$+2AP treated mice was due to a response of female mice to 2AP treatment, which was not apparent in male mice (Fig 1B and 1C). These data indicate that injections of 2AP can modulate the weight loss induced by treatment with $CCl_4$, but only in female mice.

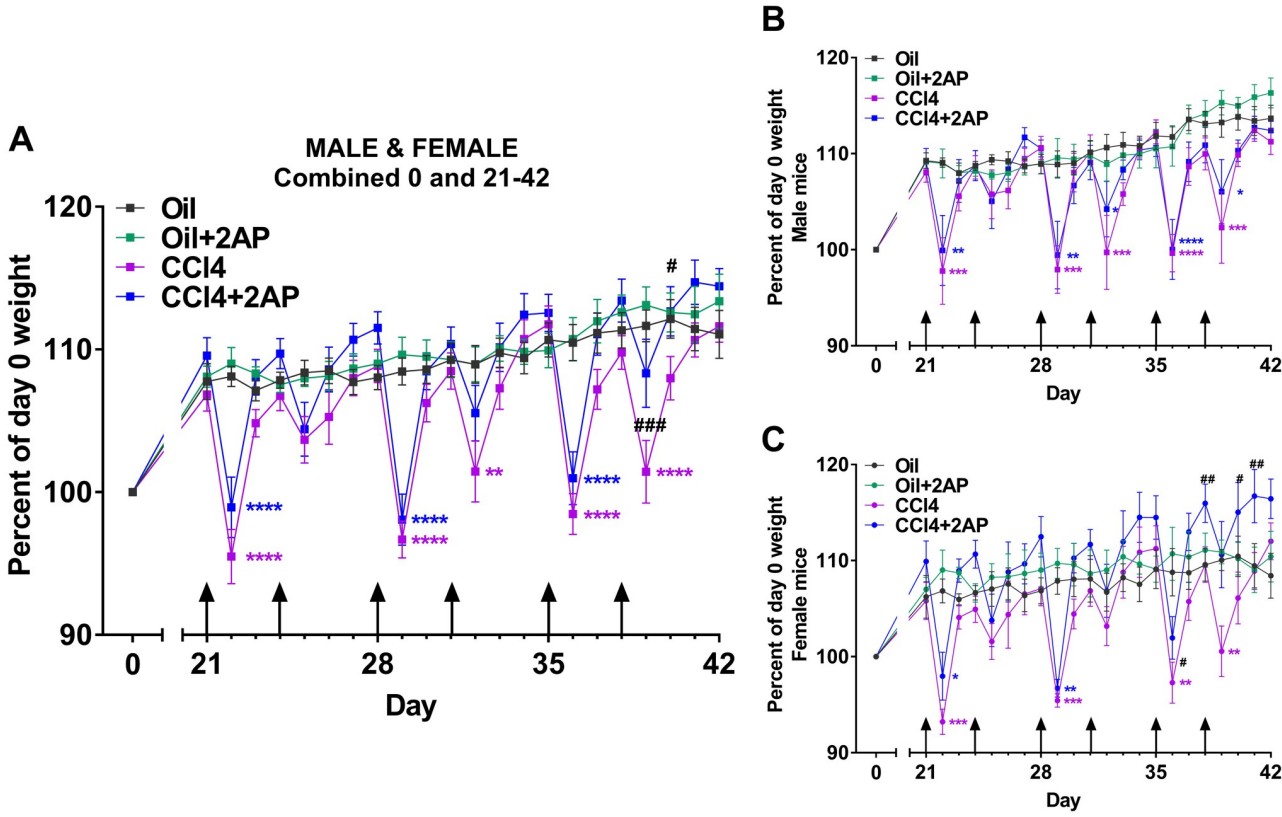

**Fig 1. Changes in body weights following CCl$_4$ and 2AP injections.** C57BL/6 male and female mice received injections of oil or CCl$_4$ in oil twice a week for 6 weeks. Starting at day 21, mice also received daily injections of 2AP or buffer control. Mice were euthanized at day 42. Graphs show body weights of **A)** male and female mice combined, **B)** male mice only, and **C)** female mice only. Values are mean ± SEM, n = 3–6 mice per group. * p < 0.05, **p < 0.01, ***p < 0.001, ****p < 0.0001 comparing control mice to mice on CCl$_4$ alone, or CCl$_4$+2AP (two-way ANOVA, Dunnett's test). # indicates p < 0.05, ## p< 0.01, and ### p<0.001 comparing mice on CCl$_4$ alone with CCl$_4$+2AP (two-way ANOVA, Dunnett's test). Arrows indicate days when mice received injections of oil or CCl$_4$.

At day 42, compared to mice that received only oil and buffer, male and female mice that received 2AP alone, CCl$_4$ alone, or CCl$_4$ with 2AP had higher liver weights (S3A Fig). Compared to our previous data [14, 22, 23, 46] and The Jackson Laboratory's Mouse Phenome Database (https://phenome.jax.org), it appears that injections of oil reduced liver weight as a percentage of total body weight. Injections of different types of oils in CCl$_4$ studies can affect liver physiology [47, 48], and components of high fat diets also modulate liver inflammation and fibrosis [49]. Injections of male mice with 2AP alone reduced white fat, and in female mice any treatment increased lung weights (S3B and S3C Fig). Injections of oil, 2AP, or CCl$_4$ had no significant effect on kidney, spleen, heart, or brown fat weights (S3B and S3C Fig). These data indicate that all mice that received CCl$_4$ had identifiable readouts of treatment (transient weight loss), and that daily injections of 2AP did not appear to have any obvious toxic effects.

## 2AP does not affect CCl$_4$-induced steatosis

Although CCl$_4$ is used as an inducer of inflammation and fibrosis in liver, CCl$_4$ can also induce steatosis [50]. CCl$_4$ is converted by the cytochrome P450 family enzyme CYP2E1 to the reactive metabolite trichloromethyl radical (CCl3•) which can then react with molecular oxygen to

form trichloromethylperoxy radical ($CCl3OO\bullet$) [51, 52]. These radicals induce mitochondrial damage and oxidative stress, and lead to hepatic lipid peroxidation and steatosis [53, 54]. 2AP had no inhibitory effect ($1.5 \pm 1.2$ percent inhibition; mean $\pm$ SD, n = 8) on the CYP450 enzymes aromatase/CYP19A1, CYP2C8, CYP1A2, CYP2B6, CYP2C19, CYP2C9, CYP2D6, and CYP3A4, or an additional 28 receptors (S2 Table).

Compared with control mice, $CCl_4$-treated mice had a significant increase in oil red O staining in both male and female mice (S4A Fig). Comparing $CCl_4$ and $CCl_4$+2AP treated male and female mice, there was no significant differences in oil red O staining in either male or female mice (S4B and S4C Fig). Compared to mice that received oil alone, mice that received oil and 2AP had no significant change in the oil red O staining (S4 Fig). These data indicate that 2AP does not significantly affect $CCl_4$-induced steatosis, and agree with our previous observation that although the general sialidase inhibitor DANA attenuates high fat diet induced steatosis, genetic loss of NEU3 does not [22].

## 2AP attenuates $CCl_4$-induced liver inflammation

Liver injury, induced either by diet, infection, or hepatotoxic drugs, leads to liver damage and inflammation [27, 55]. Compared to mice that received injections of oil alone, mice that received $CCl_4$ had an accumulation of cells in the regions around vessels of the liver (Fig 2A–2H). Although $CCl_4$ injections induced inflammation in both male and female mice (Fig 2I), the accumulation of cells around vessels was greater in the male mice (Fig 2J and 2K). Injections of 2AP attenuated the $CCl_4$ induced inflammation in both male and female mice (Fig 2). Compared to male mice, female mice that received oil had less vessel inflammation (Fig 2).

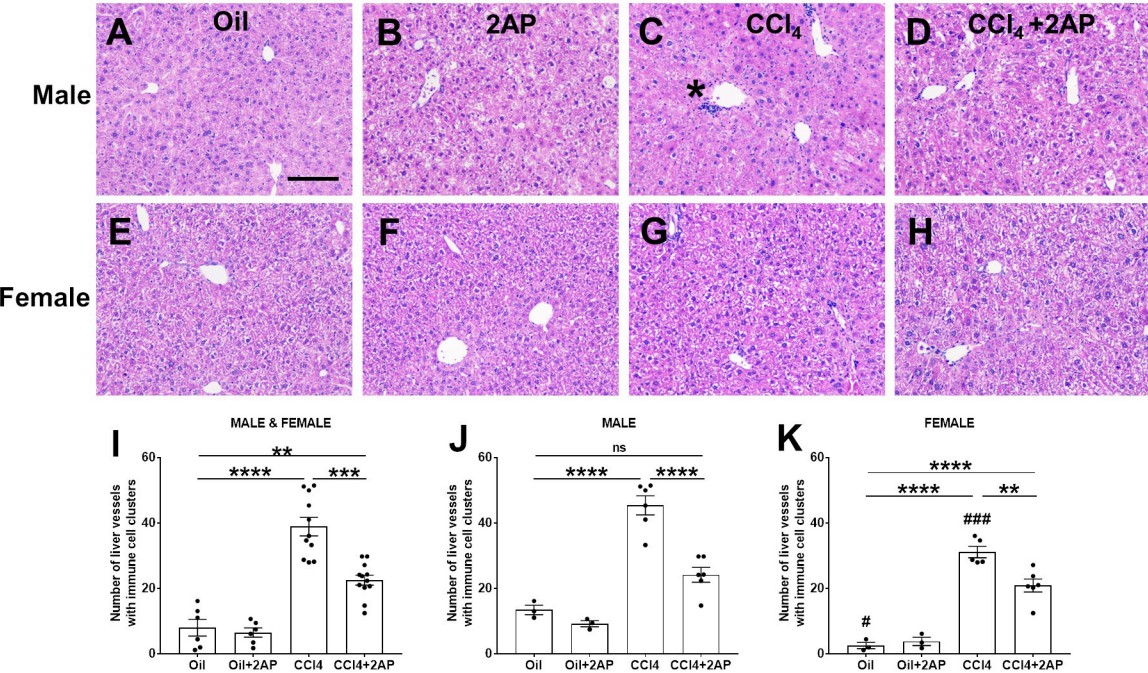

**Fig 2. 2AP injections reduce $CCl_4$-induced liver inflammation. A-H)** Representative liver sections were stained with hematoxylin and eosin. Bar is 0.1 mm. Asterisk indicates area of immune cell cluster adjacent to a vessel. **I-K)** Quantification of liver sections stained with hematoxylin and eosin. At least 100 vessels per section were assessed for the accumulation of immune cells. Values are mean $\pm$ SEM, n = 3 to 6 mice per group. **$^{**}p < 0.01$, $^{***}p < 0.001$, and $^{****}p < 0.0001$ (one-way ANOVA, Dunnett's test). ns indicates not significant. # indicates $p < 0.05$, and ### $p < 0.001$ comparing male and female mice in Fig 2J and 2K that received either oil alone or $CCl_4$ alone (one-way ANOVA, Bonferroni's test).

There were no significant differences in inflammation between male and female mice that received oil and 2AP, or CCl4 and 2AP (Fig 2).

Mouse models of liver disease are associated with increases of neutrophils and macrophages in the liver, and some but not all models show an increase of T cells [5, 27, 56]. Liver macrophages are a diverse group of cells including embryonically-derived Kupffer cells and recruited monocyte-derived macrophage subsets that may either promote or inhibit liver damage and repair [27, 57, 58]. To determine if treatment with 2AP affects these increases in immune cells, sections were stained for CLEC4F positive tissue resident liver macrophages (Kupffer cells), Mac2 (also known as Galectin-3) for inflammatory macrophages, CD3 for T cells, and S100A8/MRP8 to detect neutrophils [59–61]. Compared to control (oil) mice, there was no significant treatment-induced difference in the numbers of CLEC4F positive Kupffer cells (S5A–S5C Fig) or CD3 T cells (S5D–S5F Fig), and there were no significant differences between male and female mice. As previously observed [62–64], compared to control mice, there was an increase in the number of neutrophils in the CCl$_4$ treated mice, but this increase was only significant in females (Fig 3A–3C and S6 Fig). Comparing CCl$_4$ and CCl$_4$+2AP treated mice, there was a reduction in the number of S100A8/MRP8 positive cells in the CCl$_4$+2AP treated female, but not male mice (Fig 3A–3C and S6 Fig). Mice that received oil alone or oil and 2AP had no significant differences in the number of S100A8 positive cells (Fig 3A–3C).

As previously observed [60, 65], compared to control mice, there was an increase in the number of Mac2 positive cells in the CCl$_4$-treated mice, and this was observed in both male and female mice (Fig 3D–3F). 2AP decreased the number of Mac2 positive cells in CCl$_4$-treated male and female mice (Fig 3D–3F). Mice that received oil alone or oil and 2AP alone had no significant differences in the number of Mac2 positive cells (Fig 3D–3F).

CCl4-induced liver inflammation is associated with the accumulation of immune cells around blood vessels (perivascular inflammation) [33, 66]. To determine if the reduction in Mac2 positive cells after 2AP injections was due to changes in the number of Mac2 cells in the tissue parenchyma or around blood vessels, we counted Mac2 positive cells in liver tissue excluding vessels (parenchymal areas), as well as areas around the vessels. Compared to control mice, there were no significant treatment-induced difference in the numbers of Mac2 positive cells in the parenchymal areas of the liver, and there were no significant differences between male and female mice (Fig 3G–3I and S7 Fig). Compared to control mice, there was an increase in the number of Mac2 positive cells around the vessels of CCl$_4$-treated male and female mice (Figs 3J–3L and 4). 2AP decreased the number of Mac2 positive cells in the vessel areas of CCl$_4$-treated male and female mice (Figs 3J–3L and 4). Mice that received oil alone or oil and 2AP alone had no significant differences in the number of Mac2 positive cells in vessel areas (Figs 3J–3L and 4). These data indicate that 2AP appears to reduce the CCl$_4$-induced accumulation of Mac2 positive cells around the vessels of the liver.

## 2AP attenuates CCl$_4$-induced liver fibrosis

Liver sections were stained with picrosirius red to detect collagen. As previously observed [67], CCl$_4$ increased picrosirius red staining in male and female mouse livers (Fig 5A–5H). 2AP decreased picrosirius red staining in the livers of CCl$_4$-treated male and female mice (Fig 5I–5K). In the oil-only controls, females had lower levels of picrosirius red staining (Fig 5J–5K). Liver tissue was also assessed for collagen content with hydroxyproline. Compared to control mice, there was an increase in hydroxyproline levels in CCl$_4$-treated mice, but this effect was only significant in male mice (S8A–S8C Fig). 2AP reduced hydroxyproline levels in the CCl$_4$-treated male mice (S8A–S8C Fig). Mice that received oil alone or oil and 2AP had no

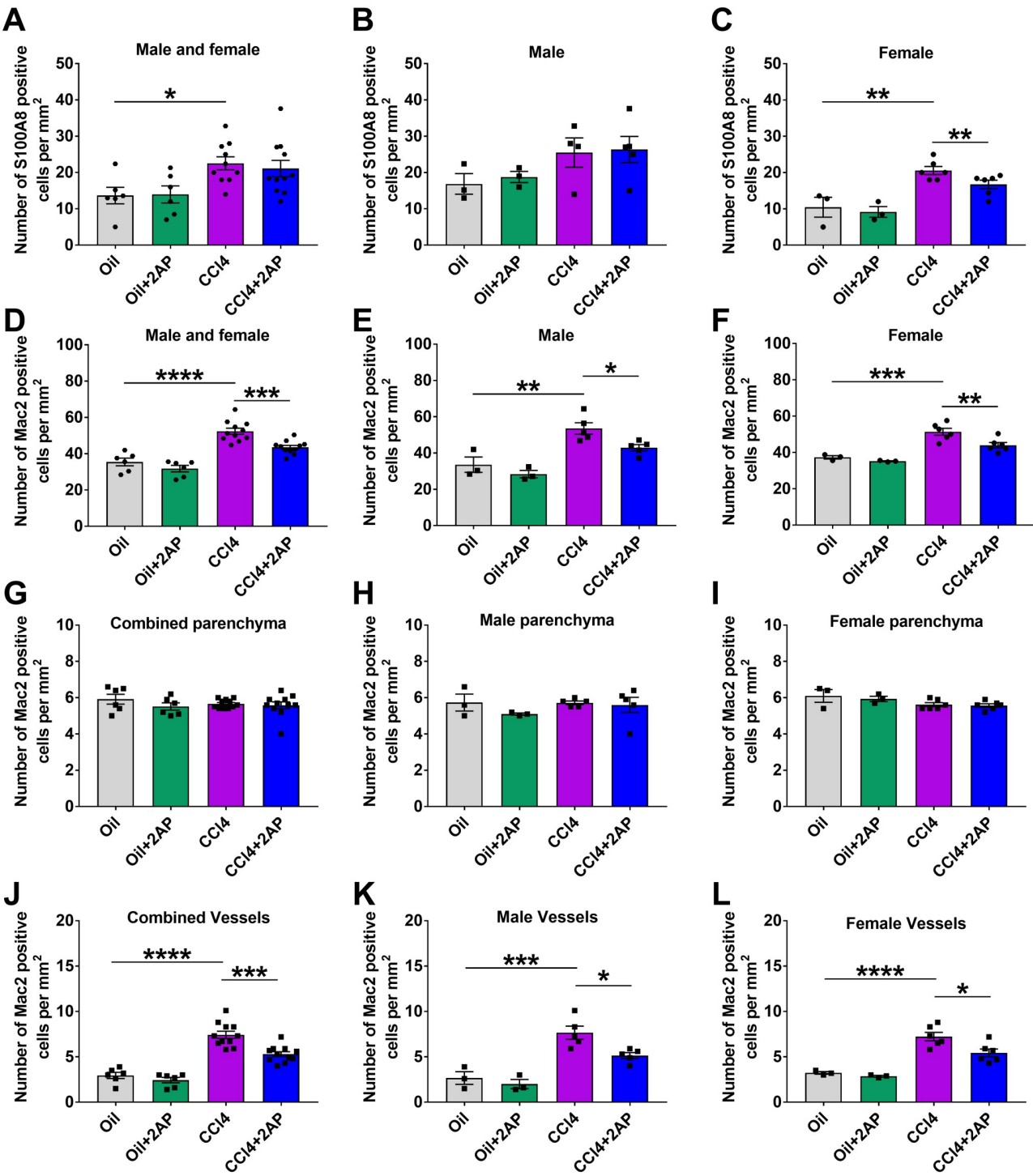

**Fig 3. 2AP injections reduce CCl₄-induced changes in liver immune cells.** Quantification of liver sections stained with antibodies for **A-C)** S100A8, **D-F)** Mac2, **G-I)** Mac2 in liver parenchyma, **J-L)** Mac2 positive cells around vessels. Values are mean ± SEM, n = 3 to 6 mice per group. *p < 0.05, **p < 0.01, ***p < 0.001, and ****p < 0.0001 (one-way ANOVA, Dunnett's test).

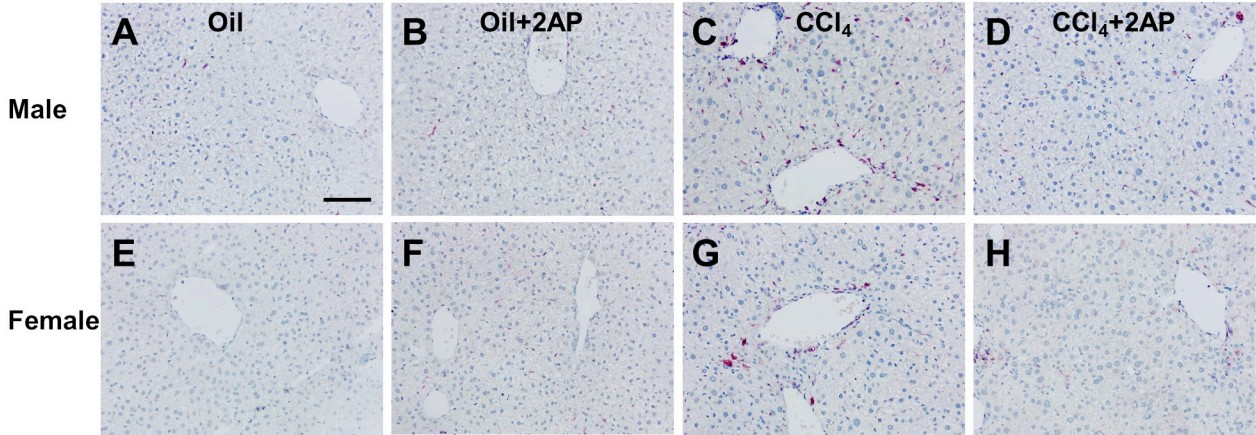

**Fig 4. 2AP injections reduce CCl$_4$ -induced increases in Mac2 positive cells.** Representative liver sections of **A-D)** male and **E-H)** female mice were stained with anti-Mac2 antibodies. Bar is 0.1 mm.

significant differences in hydroxyproline levels (S8A–S8C Fig). To determine if there was a correlation between inflammation and fibrosis, we plotted the number of liver blood vessels with immune cell clusters against the percent area stained with picrosirius red. We observed a correlation between inflammation and fibrosis in both male and female mice (S8D–S8F Fig). These data suggest that, as expected [5, 55, 56], there is a link between inflammation and fibrosis in liver disease.

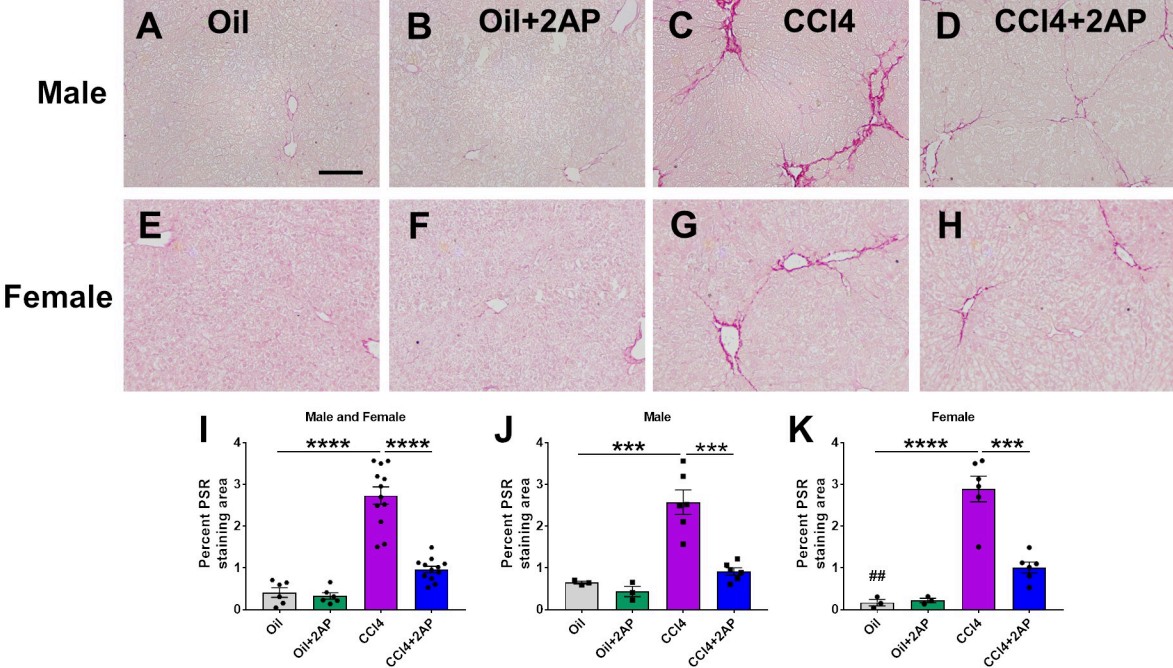

**Fig 5. 2AP injections reduce CCl$_4$ –induced changes in liver fibrosis.** Representative liver sections of **A-D)** male and **E-H)** female mice were stained with picrosirius red. Bar is 0.1 mm. **I-K)** Quantification of liver sections stained with picrosirius red (PSR). Values are mean ± SEM. n = 3 to 6 mice per group. ***$p < 0.001$, and ****$p < 0.0001$ (one-way ANOVA, Dunnett's test). ## indicates $p < 0.01$ comparing male and female mice on oil alone (t-test).

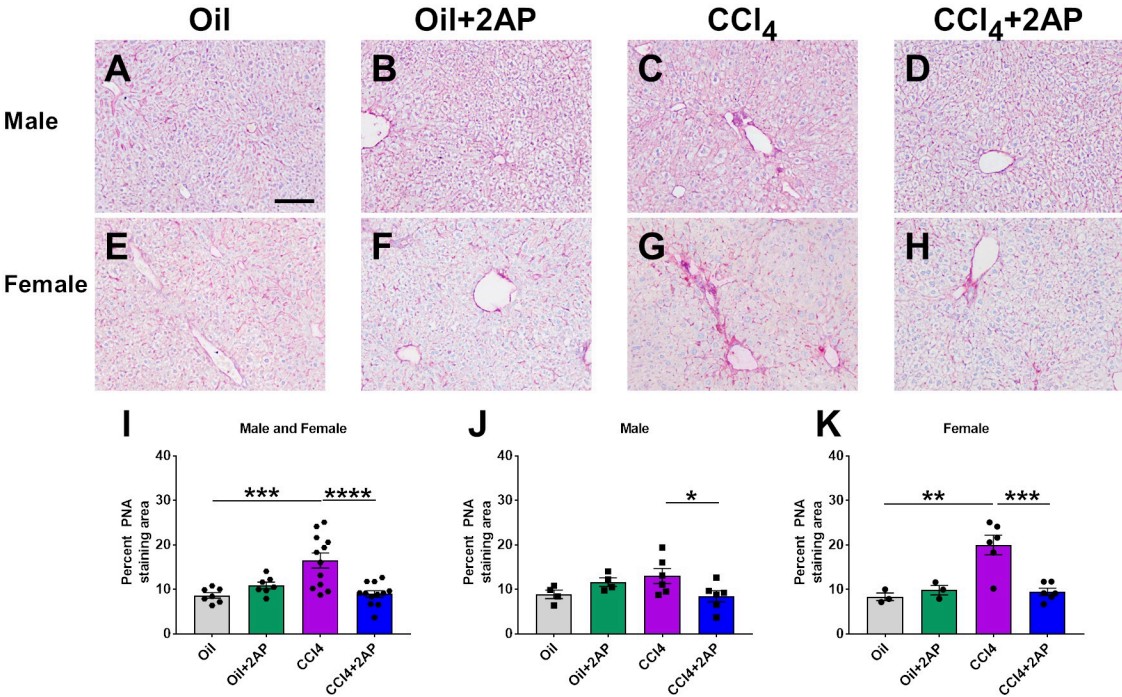

**Fig 6. 2AP injections reduce CCl₄ -induced desialylation.** Representative liver sections of **A-D)** male and **E-H)** female were stained with PNA lectin. Nuclei are counterstained blue with hematoxylin. Bar is 0.1 mm. **I-K)** Quantification of liver sections stained with PNA lectin. Values are mean ± SEM, n = 3 to 6 mice per group. *p < 0.05, **p < 0.01, ***p < 0.001, and ****p < 0.0001 (one-way ANOVA, Dunnett's test).

## 2AP attenuates CCl₄-induced liver desialylation

Loss of terminal sialic acid residues from glycan chains (desialylation) is common in many pathogenic processes, including inflammation and fibrosis [9, 68]. The lectin PNA binds to desialylated glycan chains [69]. Inhibiting inflammation and fibrosis with the general sialidase inhibitor DANA reversed desialylation in lung fibrosis [14–16]. Compared to control mice, CCl₄- treated mouse livers had increased staining with PNA in female mice (Fig 6A–6K) 2AP reduced PNA staining in the CCl₄-treated male and female mice (Fig 6A–6K). Compared to oil alone, 2AP and oil did not significantly change PNA staining (Fig 6A–6K). These data suggest that CCl₄-treated female mice have increased desialylation, and that 2AP decreases desialylation in CCl₄-treated mice.

## Elevation of NEU3 in CCl₄-induced liver disease

Sialidases are increased during lung inflammation and fibrosis in multiple animal models, and inhibiting sialidases attenuates these processes [16, 22, 23, 30]. To determine whether CCl₄ injections lead to changes in NEU3 levels, we stained sections of liver tissue for NEU3. Compared to mice that received injections of oil alone, male and female mice that received CCl₄ had an increase in NEU3 staining (Fig 7A–7H). Compared to control and F4/80 antibody staining (S9 Fig), the staining of NEU3 was present in both the nucleus and cytoplasm in the large numerous cells (hepatocytes), but the liver Kupffer (small spindle-shaped cells) and hepatic stellate cells (small stellate-shaped cells) did not stain with NEU3 antibodies (Fig 7). 2AP attenuated the CCl₄-induced NEU3 staining in both male and female mice (Fig 7I–7K).

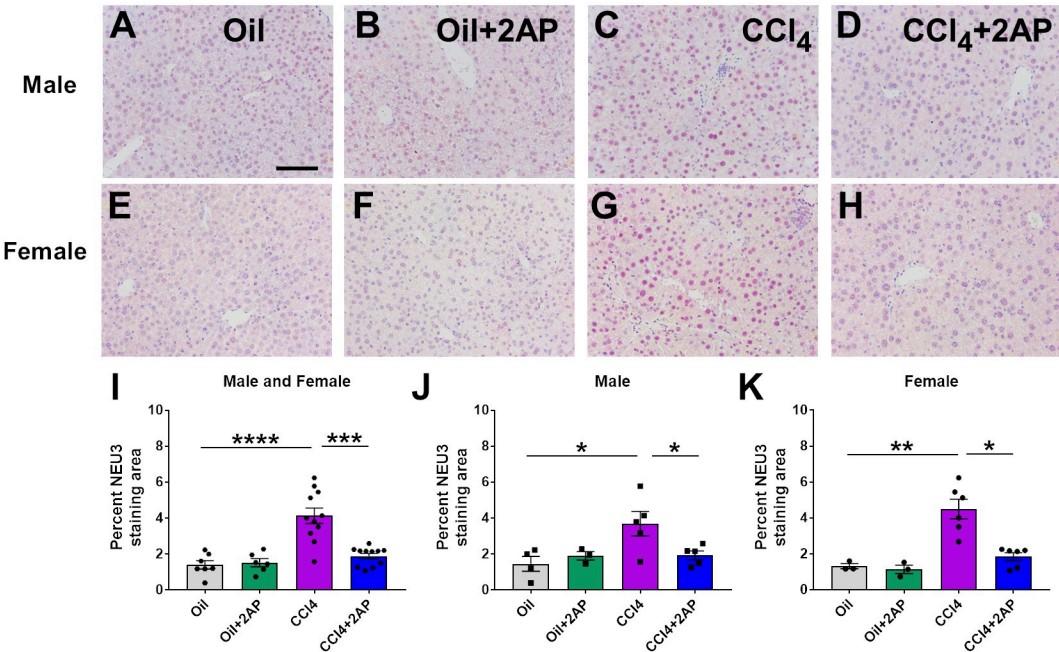

**Fig 7. 2AP injections reduce NEU3 levels in CCl$_4$ -treated mice.** Representative liver sections of **A-D)** male and **E-H)** female mice were stained for NEU3. Bar is 0.1 mm. **I-K)** Quantification of NEU3 staining. Five randomly chosen fields of view were imaged, and the percent area of the tissue showing staining was measured and the average was calculated. Values are mean ± SEM, n = 3 to 6 mice per group. *p < 0.05, **p < 0.01, ***p < 0.001, and ****p < 0.0001 (one-way ANOVA, Dunnett's test).

Mice that received oil alone or 2AP and oil alone had no significant differences in NEU3 staining (Fig 7).

## Serum from NASH patients contain a more desialylated protein

We previously observed that patients with pulmonary fibrosis have increased desialylation of serum proteins [40]. To determine whether there is abnormal sialylation of serum components in patients with liver disease (S1 Table), western blots of patient sera were stained with RCA lectin, which detects non-sialylated glycoconjugates [69, 70]. Compared to control sera, sera from NAFLD and NASH patients had increased desialylation of a ~250 kDa protein, with higher levels of desialylation in NASH patients (Fig 8A and 8B, and S1 Raw image). There was no significant difference in desialylation of other serum proteins. Western blots of sera, stained for NEU3, showed a single band at ~ 48 kDa consistent with NEU3 protein (Fig 8C and S1 Raw image), with an increase in levels of NEU3 in NAFLD patients, but not in NASH patients (Fig 8D). These data indicate that chronic liver disease may lead to increased serum NEU3 levels and increased desialylation of a serum protein.

## Discussion

In this report, we observed that 2AP attenuated CCl$_4$-induced liver inflammation and fibrosis in both male and female mice. CCl$_4$-induced liver desialylation and increased expression of NEU3 were also reversed by 2AP. The sera from patients with NAFLD and NASH also had increased levels of a desialylated serum protein and patients with NAFLD had increased serum

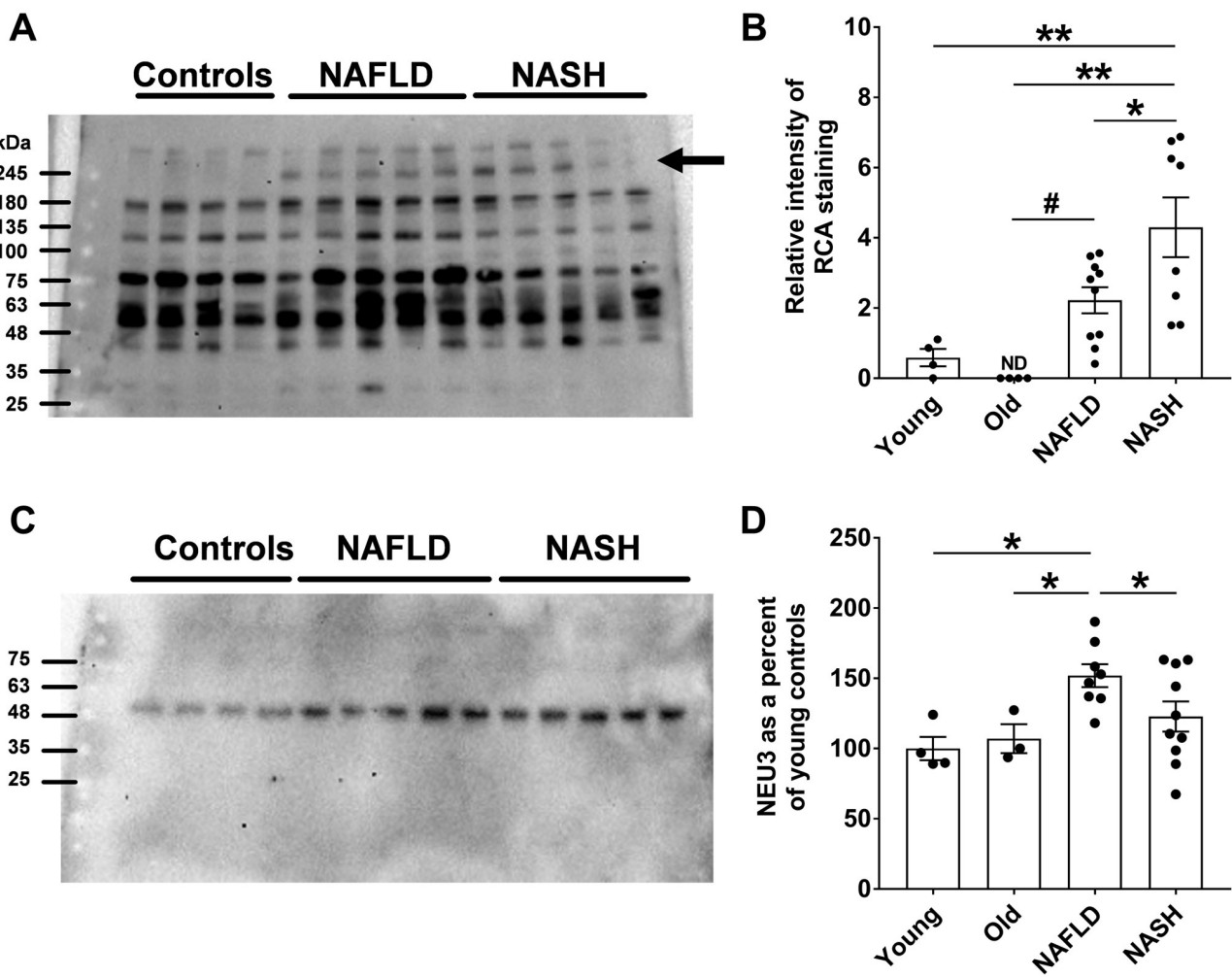

**Fig 8. Sera from NAFLD patients show elevated desialylation of a glycoprotein and elevated levels of NEU3. A)** A western blot of serum samples from young and old healthy controls, NAFLD patients, and NASH patients was stained with RCA lectin. **B)** The lectin-stained ~ 250 kDa band (indicated by the arrow in **A**) was measured by densitometry, normalizing for each sample the intensity of the 250 kDa band to the integrated intensity of the lane. **C)** A western blot of human serum samples was stained with anti-NEU3 antibodies. **D)** Quantification of NEU3. Values are mean ± SEM. *p < 0.05, **p < 0.01 (one-way ANOVA, Holm-Šídák test). # p < 0.05 (t test).

levels of NEU3. These data suggest that increased NEU3 may be associated with liver inflammation and fibrosis and that a NEU3 inhibitor can, in part, reverse these effects.

We have previously shown that DANA, or the lack of NEU3, both attenuate diet-induced liver inflammation and fibrosis in mice [22, 23]. However, DANA also inhibited diet-induced weight gain and reduced liver steatosis, whereas *NEU3*⁻/⁻ knock out mice were not resistant to diet-induced weight gain and steatosis [22, 23]. In this report, we show that 2AP, a potentially specific NEU3 inhibitor [14], did not modulate steatosis, and had only a modest effect on weight change in female mice. These data suggest that other sialidases, but not NEU3, are probably involved in regulating diet induced weight changes and steatosis.

It is unlikely that 2AP attenuates inflammation and fibrosis by inhibiting CYP2E1 for three reasons. First, the structure of sialidases and CYP450 enzymes are different, with the active site of sialidases being comprised of a group of 10 mainly tyrosine and arginine residues, whereas CYP450 enzymes are hemoproteins with a heme-iron center at the active site [71, 72].

Secondly, the structures of NEU3 and CYP2E1 inhibitors are different, with sialidase inhibitors based on pyridine and sialic acid structures, and CYP2E1 inhibitors based on imidazole, triazole, and omega fatty acid compounds [14, 52, 73]. Finally, 2AP had no inhibitory effect on 8 CYP450 enzymes or 28 receptors, including adenosine, adrenergic, dopamine, and muscarinic receptors, all of which are involved in liver inflammation and fibrosis [74–77].

CCl$_4$-induced liver injury leads to inflammation [27, 50, 55, 67]. 2AP inhibited aspects of CCl$_4$-induced inflammation in both male and female mice, such as the increased number of Mac2 cells adjacent to liver vessels. Desialylation of liver tissue, as indicated by PNA lectin staining, was increased in CCl$_4$-treated mice, especially around the vessels, and 2AP reversed this effect. These data suggest that the CCl$_4$-induced inflammatory response either increases recruitment of Mac2 cells into the liver, or inhibits their egress. Whether this is caused by elevated levels of desialylated proteins around the vessels is unclear.

Immune cells can migrate into the liver from both the hepatic artery and hepatic portal vein and travel through the liver sinusoids where they interact with the sinusoidal endothelial cells, Kupffer cells, and hepatic stellate cells, and if activated, the immune cells can enter the liver tissue [78, 79]. Following an immune response, inflammatory cells may leave the liver via the lymphatic system or the central vein by processes variously called "reverse migration", "retrograde chemotaxis", or "chemorepulsion" [4, 80–83]. Sialidases can modulate cell adhesion and migration, and sialidase inhibitors attenuate these events [84–87]. Although all 4 sialidases are found in the cytoplasm, NEU3 is also expressed on the extracellular membrane [10, 11], and NEU3 is also found in serum and bronchoalveolar lavage fluid [15, 16, 40]. We observed that after CCl$_4$ treatment, hepatocytes increased expression of both cytoplasmic and nuclear NEU3, as observed previously for epithelial cells, CHO cells, and glioblastoma cells [11, 88, 89]. How the increased expression of NEU3 in the liver, especially in the hepatocytes, leads to increased desialylation of glycoconjugates associated with the vessels is unclear. One possibility is that the natural flow from the liver portal triad to the central vein would also carry extracellular NEU3 towards the central vein [4, 5, 78]. Whether our observation of the CCl$_4$-induced accumulation of Mac2 cells around vessels and the reversal of this effect by 2AP is part of this process is unclear.

There is increased prevalence of liver disease in men compared to women, and in experimental models male mice have a more pronounced inflammatory response to liver injury than female mice, which may in part be due to sex hormones [25, 28, 29]. Although neutrophils exacerbate acute liver injury, increased numbers of neutrophils in chronic liver injury models may lead to less inflammation and fibrosis and enhanced recovery [5, 62, 63, 90, 91]. We also found that, compared to male mice, CCl$_4$-treated female mice had an increase in S100A8/MRP8 positive neutrophils, and that 2AP reversed this increase. Whether the increase in neutrophils in female compared to male mice is linked to the more pronounced inflammatory response in male mice is unclear [27–29, 50, 55, 67].

Upregulation of NEU3 protein and increased desialylation of serum proteins is associated with multiple diseases including intestinal inflammation and colitis, rheumatoid arthritis, neuroinflammation, heart disease, and lung fibrosis [9, 12–16, 92–94]. We observed that 2AP inhibited the CCl$_4$-induced upregulation of NEU3. NEU3 protein production is promoted by both TGF-β1 and hypoxia signaling pathways. A NEU3—TGF-β1 positive feedback loop is regulated by the RNA-binding protein DDX3, and the transcription factor hypoxia-inducible factor 2α (HIF-2α) regulates NEU3 in a HIF-2α–NEU3–ceramide pathway [14, 17, 46, 95]. Hepatocytes and HSCs are exposed to TGF-β1 and hypoxia in liver disease, and these events lead to dysregulated metabolism and release inflammatory signals, which then perpetuate liver inflammation and drive fibrosis [3, 96].

Increased levels of NEU3 associated with inflammation and fibrosis correlate with increased desialylation of some serum proteins [9, 12–16]. We observed that patients with liver disease also have increased desialylation of some serum proteins and elevated serum levels of NEU3 protein. These data appear to indicate that NEU3 is involved in liver disease and suggests that NEU3 inhibitors might be a novel therapeutic for liver inflammation and fibrosis.

## Conclusion

We have demonstrated that the NEU3 inhibitor 2AP attenuates $CCl_4$-induced liver inflammation, fibrosis, desialylation of proteins, and increased expression of NEU3 in both male and female mice. Together with the observation that sera from patients with NAFLD and NASH also had increased levels of a desialylated serum protein and patients with NAFLD had increased serum levels of NEU3, these findings suggest that 2AP, or other NEU3 inhibitors, should be considered as potential therapeutics for liver inflammation and fibrosis.

## Supporting information

**S1 Fig. Quantification of PNA staining in liver tissue. A-C)** Representative liver sections were stained with PNA lectin (red staining). Bar is 0.1 mm. **D-F)** Threshold masking (black) of lectin-stained areas used for quantification.
(TIF)

**S2 Fig. Changes in body weights following $CCl_4$ and 2AP injections over 42 days.** C57BL/6 male and female mice received injections of oil or $CCl_4$ in oil twice a week for 42 days. Starting at day 21, mice also received daily injections of 2AP or buffer control. Mice were euthanized at day 42. Graphs show body weights of **A)** male and female mice combined, **B)** male mice only, and **C)** female mice only. **D)** Comparison of body weight changes after $CCl_4$ injections in male and female mice from days 21 to 42. **E)** Percent weight change at 24 hours after injection of oil or $CCl_4$ in oil as a percent of weight on day of injection in male and female mice. Values are mean ± SEM, n = 3–6 mice per group. * indicates $p < 0.05$, **$p < 0.01$, ***$p < 0.001$, ****$p < 0.0001$ comparing control mice to mice on $CCl_4$ alone, or $CCl_4$+2AP (two-way ANOVA, Dunnett's test). # indicates $p < 0.05$, ## $p < 0.01$, and ### $p < 0.001$ comparing mice on $CCl_4$ alone with $CCl_4$+2AP (two-way ANOVA, Dunnett's test). **E)** *** $p < 0.001$ (one-way ANOVA, Dunnett's test). Arrows indicate days when mice received injections of oil or $CCl_4$.
(TIF)

**S3 Fig. Changes in organ weights following $CCl_4$ and 2AP injections.** C57BL/6 mice were treated as in Fig 1 and euthanized at day 42. **A)** Liver weights from male and female mice combined, **B)** organ weights from male mice only, and **C)** organ weights from female mice only. Values are mean ± SEM, n = 3–6 mice per group. * $p < 0.05$, **$p < 0.01$, ***$p < 0.001$ (one-way ANOVA, Dunnett's test).
(TIF)

**S4 Fig. Injections of 2AP do not reduce $CCl_4$ -induced liver steatosis. A-C)** Quantification of liver sections stained with oil red O (ORO). Values are mean ± SEM, n = 3 to 6 mice per group. **$p < 0.01$ and ****$p < 0.0001$ (one-way ANOVA, Dunnett's test).
(TIF)

**S5 Fig. Injections of $CCl_4$ do not alter Kupffer or CD3 T cells.** Quantification of liver sections stained with antibodies for **A-C)** CLEC4F-positive Kupffer cells, **D-F)** CD3-positive T cells. Values are mean ± SEM, n = 3 to 6 mice per group.
(TIF)

**S6 Fig. 2AP injections reduce CCl₄ -induced increases in S100A8/MRP8 positive cells.** Representative liver sections of **A-D)** male and **E-H)** female mice were stained with anti- MRP8 antibodies. Bar is 0.1 mm.
(TIF)

**S7 Fig. Mac2 staining in liver parenchyma.** Representative liver parenchyma sections of **A-D)** male and **E-H)** female mice were stained with anti-Mac2 antibodies. Bar is 0.1 mm.
(TIF)

**S8 Fig. 2AP injections reduce CCl₄ -induced changes in liver hydroxyproline levels and correlation of inflammation and fibrosis. A-C)** Quantification of liver tissue for hydroxyproline (μg/mg liver tissue). Values are mean ± SEM, n = 3 to 6 mice per group. *$p < 0.05$ (one-way ANOVA, Dunnett's test). **D-F)** Correlation of the number of liver blood vessels with immune cell clusters with percent picrosirius red staining.
(TIF)

**S9 Fig. Control antibody staining of liver sections.** Sections of livers from **A-C)** oil alone control or **D)** CCl₄ injected mice were stained with **A)** no primary antibodies, **B)** anti-F4/80 antibodies, or **C)** and **D)** anti-NEU3 antibodies. Bar is 0.1 mm.
(TIF)

**S1 Appendix. IRB approval letters related to this study.**
(7Z)

**S1 Table. Clinical details of patients.** Clinical data from the National Institute of Diabetes and Digestive and Kidney Diseases Central Repository (NIDDK-CR).
(DOCX)

**S2 Table. CYP450 enzymes and receptor inhibition by 2AP.** CYP450 enzymes and a panel of receptors were assessed for inhibition by 10 μM 2AP.
(DOCX)

**S1 Raw image. Raw images of western blots for Fig 8.** "Sera from NAFLD patients show elevated desialylation of a glycoprotein and elevated levels of NEU3". **A)** Whole raw blot image of western blots of serum samples from young and old healthy controls, NAFLD patients, and NASH patients was stained with RCA lectin. **B)** Whole raw blot images of western blots of human serum samples was stained with anti-NEU3 antibodies.
(TIF)

## Acknowledgments

We thank the LARR (Laboratory Animal Resources and Research) staff at Texas A&M University for animal care. The authors acknowledge the assistance of the TAMU College of Veterinary Medicine Core Histology Lab Research Unit (RRID:SCR_022201). We also thank Sumeen Gill for helpful comments on the manuscript. We are grateful to Dr. Erica Herzog, Yale University Section of Pulmonary, Critical Care, and Sleep Medicine for providing old control serum samples.

## Author Contributions

**Conceptualization:** Darrell Pilling, Richard H. Gomer.

**Data curation:** Darrell Pilling.

**Formal analysis:** Darrell Pilling, Trevor C. Martinez, Richard H. Gomer.

**Investigation:** Trevor C. Martinez, Richard H. Gomer.

**Visualization:** Darrell Pilling, Trevor C. Martinez, Richard H. Gomer.

**Writing – original draft:** Darrell Pilling, Richard H. Gomer.

**Writing – review & editing:** Darrell Pilling, Trevor C. Martinez, Richard H. Gomer.

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
