## [Decision Letter · Decision Letter 0]

16 Sep 2024

PONE-D-24-29550Inhibition of CCl4-induced liver inflammation and fibrosis by a NEU3 inhibitorPLOS ONE

Dear Dr. Gomer,

Thank you for submitting your manuscript to PLOS ONE. After careful consideration, we feel that it has merit but does not fully meet PLOS ONE’s publication criteria as it currently stands. Therefore, we invite you to submit a revised version of the manuscript that addresses the points raised by the reviewers.

We look forward to receiving your revised manuscript.

Kind regards,

Matias A Avila, Ph.D.

Academic Editor

PLOS ONE

2. We note that you have a patent relating to material pertinent to this article. Please provide an amended statement of Competing Interests to declare this patent (with details including name and number), along with any other relevant declarations relating to employment, consultancy, patents, products in development or modified products etc. Please confirm that this does not alter your adherence to all PLOS ONE policies on sharing data and materials, as detailed online in our guide for authors http://journals.plos.org/plosone/s/competing-interests by including the following statement: "This does not alter our adherence to  PLOS ONE policies on sharing data and materials.” If there are restrictions on sharing of data and/or materials, please state these. Please note that we cannot proceed with consideration of your article until this information has been declared.

Reviewers' comments:

Reviewer's Responses to Questions

**Comments to the Author**

1. Is the manuscript technically sound, and do the data support the conclusions?

Reviewer #1: Partly

Reviewer #2: Yes

2. Has the statistical analysis been performed appropriately and rigorously? 

Reviewer #1: Yes

Reviewer #2: Yes

3. Have the authors made all data underlying the findings in their manuscript fully available?

Reviewer #1: Yes

Reviewer #2: Yes

4. Is the manuscript presented in an intelligible fashion and written in standard English?

Reviewer #1: No

Reviewer #2: Yes

5. Review Comments to the Author

Reviewer #1: This study was performed to demonstrate the effect of 2-acetyl pyridine (2AP), an inhibitor of neuraminidase 3 (NEU3), on the liver inflammation and fibrosis experimentally induced by CCl4. Results are of interest; however, some issues still need to addressed before further steps in the publication process.

The manuscript is too long, it contains repeated information in the introductory paragraph of results section and in the description of discussion section, and authors cited 116 references. So, an original manuscript should be limited to no more than 50 to 60 references. This manuscript can easily be shortened by 30% if authors reorganize its content.

Representative pictures of IHC analyses (Fig 3) of anti-S100A8 and anti-Mac2 should be included as evidence of where the quantifications came from.

In this document is missing a strong conclusion at the end of the discussion section; instead, authors included a kind of incomplete conclusion at the beginning of discussion section: “In this report, we observed that 2AP attenuated CCl4-induced liver inflammation and fibrosis in both male and female mice. CCl4-induced liver desialylation and increased expression of NEU3 were also reversed by 2AP. The sera from patients with NAFLD and NASH also had increased levels of a desialylated serum protein and patients with NAFLD had increased serum levels of NEU3. These data suggest that increased NEU3 may be associated with liver inflammation and fibrosis and that a NEU3 inhibitor can, in part, reverse these effects”.

The clinical relevance of 2AP on liver fibrosis should be clearly stated in the conclusion section of the manuscript.

This manuscript contains some semantic and grammar issues that need to be carefully corrected; e.g., in results section, authors exaggerate to use “compare to”, “there was/were”, etc.

Reviewer #2: This manuscript is well-written and describes a set of rigorously performed experiments demonstrating that the NEU3 inhibitor, 2AP, attenuates development of CCl4-induced liver inflammation and fibrosis. Experiments were conducted in both male and female mice and data were carefully analyzed in aggregate as well as separately. Appropriate controls were performed, including controls administering vehicle and inhibitor alone. NEU3 activity was also demonstrated in patient samples, increasing the clinical relevance of the paper’s findings. This reviewer has only minor additional comments:

1) Can the authors speculate on the significance of the acute reversible weight loss in mice immediately after each CCl4 injection? In addition, what may be the reason for the lack of acute weight loss in female mice at day 40 treated with 2AP?

2) For Figure 2, please clarify the total number of liver vessels that were evaluated in each sample that generated the reported numbers of vessels with inflammatory cells reported in the quantification graphs I-K.

3) Please explain why inhibitory activity of 2AP on CYP2E1 function was not directly tested. Are CYP2E1 functional assays not available and/or difficult to perform?

6. PLOS authors have the option to publish the peer review history of their article (what does this mean?). If published, this will include your full peer review and any attached files.

Reviewer #1: No

Reviewer #2: No

---

## [Author Response · Author response to Decision Letter 0]

25 Sep 2024

Matias A Avila, Ph.D.

Academic Editor

PLOS ONE September 23, 2024

Dear Dr. Avila,

Thank you for considering our manuscript PONE-D-24-29550 “Inhibition of CCl4-induced liver inflammation and fibrosis by a NEU3 inhibitor” for publication in PLOS ONE. 

As detailed below, we have carefully responded to the suggestions from the reviewers and have included additional images related to the analysis of MRP8/S100A8 positive cells in the liver (Figure S6) and the raw whole blot images related to the analysis of desialylation of proteins and levels of NEU3 in patient sera (Figure S10). As suggested by reviewer 1, we have also reduced the number of references in the manuscript and added a conclusion section. Following the comments from reviewer 2 we have clarified the methods section regarding assessment of vessel inflammation. 

We have also uploaded a revised marked-up copy of the manuscript with track changes, an unmarked revised manuscript, and provided an updated competing interests and financial disclosure sections.

We confirm that the manuscript does conform to PLOS ONE style requirements.

2. We note that you have a patent relating to material pertinent to this article. Please provide an amended statement of Competing Interests to declare this patent (with details including name and number), along with any other relevant declarations relating to employment, consultancy, patents, products in development or modified products etc. Please confirm that this does not alter your adherence to all PLOS ONE policies on sharing data and materials, as detailed online in our guide for authors by including the following statement: 

"This does not alter our adherence to PLOS ONE policies on sharing data and materials.” If there are restrictions on sharing of data and/or materials, please state these. Please note that we cannot proceed with consideration of your article until this information has been declared.

We confirm that all requirements related to financial and competing interest statements have been completed.

We confirm that the raw whole blot images related to the analysis of desialylation of proteins and levels of NEU3 in patient sera have been included in the manuscript (Figure S10).

While revising your submission, please upload your figure files to the Preflight Analysis and Conversion Engine (PACE) digital diagnostic tool, https://pacev2.apexcovantage.com/. PACE helps ensure that figures meet PLOS requirements.

We confirm that Figures 1-8 conform to PACE specifications. 

Review Comments to the Author

Reviewer #1: This study was performed to demonstrate the effect of 2-acetyl pyridine (2AP), an inhibitor of neuraminidase 3 (NEU3), on the liver inflammation and fibrosis experimentally induced by CCl4. Results are of interest; however, some issues still need to addressed before further steps in the publication process.

The manuscript is too long, it contains repeated information in the introductory paragraph of results section and in the description of discussion section, and authors cited 116 references. So, an original manuscript should be limited to no more than 50 to 60 references. This manuscript can easily be shortened by 30% if authors reorganize its content.

We have reduced the number of references by 20 and have shortened the manuscript as much as possible. However, we respectfully disagree that we should reduce our reference list to 50-60. If we reduced the number of references to 60 then all references related to the methods section would have to be removed, and the number of references related to previous work and interpretation of the results would be scant. In addition, the lack of references would make it appear that we do not appreciate previous work in this area of research and it would reduce our ability to interpret our findings with regard to previously published works. This same reasoning is why we did not reduce the manuscript length by 30%. One of the major benefits of publishing in PLOS ONE is that the journal allows space to provide a detailed method section (with relevant references) and allows for a careful explanation of the results and a detailed discussion.

Representative pictures of IHC analyses (Fig 3) of anti-S100A8 and anti-Mac2 should be included as evidence of where the quantifications came from.

We have now added an additional figure (Figure S6) showing anti-S100A8 staining. 

For anti-Mac2 staining, we believe that Figure 4 (Fig 4. 2AP injections reduce CCl4 -induced increases in Mac2 positive cells) and Figure S6 (S6 Fig. Mac2 staining in liver parenchyma), (now S7 Fig in the revised manuscript) provide representative images of anti-Mac2 staining. 

In this document is missing a strong conclusion at the end of the discussion section; instead, authors included a kind of incomplete conclusion at the beginning of discussion section: “In this report, we observed that 2AP attenuated CCl4-induced liver inflammation and fibrosis in both male and female mice. CCl4-induced liver desialylation and increased expression of NEU3 were also reversed by 2AP. The sera from patients with NAFLD and NASH also had increased levels of a desialylated serum protein and patients with NAFLD had increased serum levels of NEU3. These data suggest that increased NEU3 may be associated with liver inflammation and fibrosis and that a NEU3 inhibitor can, in part, reverse these effects”.

The clinical relevance of 2AP on liver fibrosis should be clearly stated in the conclusion section of the manuscript.

We have now added a conclusion section to the manuscript discussing possible clinical relevance of 2AP on liver fibrosis (lines 562-568 of the revised manuscript with track changes); thank you for this suggestion!

This manuscript contains some semantic and grammar issues that need to be carefully corrected; e.g., in results section, authors exaggerate to use “compare to”, “there was/were”, etc.

We respectfully disagree with reviewer 1 and believe that our use of the terms indicated above are appropriate.

Reviewer #2: This manuscript is well-written and describes a set of rigorously performed experiments demonstrating that the NEU3 inhibitor, 2AP, attenuates development of CCl4-induced liver inflammation and fibrosis. Experiments were conducted in both male and female mice and data were carefully analyzed in aggregate as well as separately. Appropriate controls were performed, including controls administering vehicle and inhibitor alone. NEU3 activity was also demonstrated in patient samples, increasing the clinical relevance of the paper’s findings. This reviewer has only minor additional comments:

1) Can the authors speculate on the significance of the acute reversible weight loss in mice immediately after each CCl4 injection? In addition, what may be the reason for the lack of acute weight loss in female mice at day 40 treated with 2AP?

We also find this result interesting. We struggled to find papers where daily weight measurements after CCl4 injections in mice were reported. We did find some papers (Crit Rev Toxicol. 2003;33(2):105-36., Biotechnology & Biotechnological Equipment. 2015;29(6):1164-8., and Toxicology & Applied Pharmacology. 1984;75(1):1-7) which show that in rats there is weight loss at day 1 after CCl4 injections, and that in humans CCl4 exposure leads to cachexia and weight loss. We added to lines 282-283 “This acute reversible weight loss may be due to a general toxicity of CCl4.”

The lack of weight loss in the female mice at day 40 may indicate that long-term dosing of 2AP may be hepato-protective, but these data require more extensive experimentation, which is beyond the scope of the current manuscript. Thus, we did not attempt to interpret these data in this manuscript.

2) For Figure 2, please clarify the total number of liver vessels that were evaluated in each sample that generated the reported numbers of vessels with inflammatory cells reported in the quantification graphs I-K.

We clarified and expanded the methods section to indicate how the counts were performed and have also included text in the figure legend for Figure 2. We analyzed at least 100 vessels per section for the accumulation of immune cells. This is now in the methods section as lines 241-244 of the revised manuscript with track changes, and in the legend for Figure 2 as lines 348-349 of the revised manuscript with track changes.

3) Please explain why inhibitory activity of 2AP on CYP2E1 function was not directly tested. Are CYP2E1 functional assays not available and/or difficult to perform?

Although several companies sell CYP2E1 ELISA kits to measure CYP2E1 protein levels, CYP2E1 functional assays are technically very difficult to perform, and CYP2E1 functional assays are only available as part of drug screening panels from commercial CRO’s.

---

## [Editor Report · Decision Letter 1]

27 Sep 2024

Inhibition of CCl4-induced liver inflammation and fibrosis by a NEU3 inhibitor

PONE-D-24-29550R1

Dear Dr. Gomer,

We’re pleased to inform you that your manuscript has been judged scientifically suitable for publication and will be formally accepted for publication once it meets all outstanding technical requirements.

Kind regards,

Matias A Avila, Ph.D.

Academic Editor

PLOS ONE
---

## [Editor Report · Acceptance letter]

4 Oct 2024

PONE-D-24-29550R1 

PLOS ONE

Dear Dr. Gomer, 

I'm pleased to inform you that your manuscript has been deemed suitable for publication in PLOS ONE. Congratulations! Your manuscript is now being handed over to our production team.

Kind regards, 

on behalf of

Dr Matias A Avila 

Academic Editor

PLOS ONE